# ROS Signaling Mediates Directional Cell Elongation and Somatic Cell Fusion in the Red Alga *Griffithsia monilis*

**DOI:** 10.3390/cells11132124

**Published:** 2022-07-05

**Authors:** Jong-Seok Moon, Chan-Young Hong, Ji-Woong Lee, Gwang-Hoon Kim

**Affiliations:** Department of Biological Sciences, Kongju National University, Gongju 32588, Korea; jongseok11@kongju.ac.kr (J.-S.M.); nicegandy@smail.kongju.ac.kr (C.-Y.H.); ljw86@kongju.ac.kr (J.-W.L.)

**Keywords:** *Griffithsia monilis*, H_2_O_2_, ROS signal, somatic fusion, wound response

## Abstract

In many filamentous red algae, cells that die from physical damage are replaced through somatic fusion of repair cells formed from adjacent cells. We visualized ROS generation in repair cells of *Giriffthsia monilis* using DCFH-DA staining and examined the expression of the genes involved in wound healing using quantitative PCR. Repair cells elongate along the H_2_O_2_ gradient, meet at each other’s tips where the H_2_O_2_ concentration is highest, and undergo somatic fusion. No wound response occurred with ascorbic acid treatment. Conversely, H_2_O_2_ treatment induced many repair cells, leading to multiple somatic cell fusions. Diphenylene iodonium (DPI) or caffeine treatment reversibly inhibited ROS production in repair cells and blocked the progression of the wound response suggesting that ROS and calcium signaling are involved in the process. Four *G. monilis* homologues of NADPH-oxidase (*GmRBOH*s) were identified. The expression of *GmRBOH*s was upregulated upon injury, peaking 1 h post injury, and decreasing to initial levels when repair cells began to elongate. Our results suggest that ROS generated upon cell injury activates Ca^2+^ channels and upregulates the expression of *GmRBOH*s, and that H_2_O_2_ generated from repair cells mediates induced repair cell elongation leading to somatic cell fusion and filament repair.

## 1. Introduction

Wound healing is a basic attribute of a multicellular system and efficient self-repair is essential to maintain organismal integrity [1]. Reactive oxygen species (ROS) have common signaling and coordination roles in the wound responses of plant and animals [2]. The capacity of the cells to simultaneously produce and scavenge ROS, enables rapid and dynamic wound-healing responses by simply tilting the balance between cellular producing and scavenging rates [3]. The signaling effects of ROS depends on the cellular context, local concentration, and exposure time [4]. In contrast to the superoxide anion and hydroxyl radical, the less reactive hydrogen peroxide (H_2_O_2_) is involved in more delicate physiological processes such as wound responses [5]. At concentrations in the low nanomolar range, H_2_O_2_ acts as a signaling molecule which integrates with phytohormones and regulates various cellular response [6]. Another advantage of H_2_O_2_ as a signaling molecule is that it can be an auto-propagating signal that can last for a relatively long time. H_2_O_2_ could activate NADPH-oxidases in cells along the signal path that amplify the signal in a positive feedback loop [7]. However, whether and how this extracellular H_2_O_2_ (e H_2_O_2_) is sensed at the cell surface is still unknown [8].

NADPH-oxidase (respiratory burst oxidase homolog, RBOH in plants; termed NOX in animals) are the only known enzymes whose specific function is ROS generation [9]. This family of enzymes were known as plasma membrane proteins but may also be found in the vacuole, endoplasmic reticulum (ER), mitochondria, and even in nuclei [10]. While ROS are easily generated by cellular metabolic activities and triggered by environmental cues, ROS levels in the cell are finely controlled by NADPH-oxidase [11]. Diphenylene iodonium (DPI) has been regarded as a useful inhibitor to study ROS-signaling because it acts only on activated NADPH-oxidases by targeting the ROS-producing heme on the external side of the cell-membrane to form stable adducts [12]. In red algae, NADPH-oxidases were identified in several species [13,14], but only very recently did their functional properties begin to be revealed [7].

The expression of NADPH-oxidase is closely regulated by cellular calcium levels [10]. NADPH-oxidase has been reported to regulate plant cell expansion through the activation of calcium channels [15]. Calcium-mediated H_2_O_2_ generation has been extensively studied in (post)fertilization of some fucoid brown algae [16,17,18]. In fucoid algae, H_2_O_2_ and Ca^2+^ has been reported to function in a positive feedback loop as modulation of Ca^2+^ results in parallel modulation of ROS levels [19]. It was recently reported that calcium-mediated ROS signaling regulates fertilization in the red alga *Bostrychia moritziana* [7]. Structural analysis using genome data showed that all known red algal NADPH-oxidases lack calcium-binding EF-domains which are required to respond directly to calcium gradients [20]. The role of calcium signaling in regulating cellular ROS production is still largely unknown in red algae.

The physical adhesion of cells to build a multicellular unit has occurred multiple times in evolution [21,22] and is believed to be advantageous for survival [23,24]. Red algae have also independently evolved multicellularity, from simple filaments to complex thalli [25]. When cells in red algal filament are damaged, the plant is in danger of losing its upper cells. Red algal filaments have been shown to repair damage by somatic fusion of repair cells from the adjacent cells and reconnect upper and lower part of the plant [26,27,28,29]. *Griffithsia* species have been used as a model in the wound healing process because of their simple filamentous structure and giant cells [26,28,29,30,31]. Although this wound-healing process has been studied extensively, the primary signals that triggers the wound responses are unknown.

A signaling glycoprotein named rhodomorphin has been claimed to mediate the elongation and attraction of repair cells [32,33,34]. The presence of rhodomorphin was indirectly demonstrated in *Griffithsia pacifica* through the microscopic observation that specific proteins secreted from repair rhizoidal cells induce differentiation and elongation of repair apical cells [32]. Partial purification of the protein has been performed using extracts of rhizoidal cells and has shown that it is a glycoprotein with an affinity for the lectin, concanavalin A [33,34]. So far, no further purification or molecular analysis of rhodomorphin has been performed.

The aim of this investigation is to elucidate the key signals that trigger the initial wound-response, chemicals mediating the directional elongation of repair cells leading to somatic fusion, and the key genes involved in this process.

## 2. Materials and Methods

### 2.1. Algal Cultures

A gametophyte strain of *Griffithsia monilis* was provided by Prof. J. A. West in 2008 and kept in laboratory culture. The thalli were maintained in unialgal cultures in IMR medium [35] at 20 °C in a 16 h light ⁄ 8 h dark cycle with illumination of >20 µmol photons m^−2^∙s^−1^ provided by cool-white fluorescent lighting. The plants were transferred every 1–2 weeks to new IMR medium.

### 2.2. Time-Lapse Observation of Wound Response

Subcultures of 10–20 filament tips approximately 1 cm long were transferred to fresh medium in 100 mL dishes a week before use. After killing a cell with a sharp razor blade, the cytoplasm was removed by gentle squeezing with a brush leaving a transparent cell wall, and the filaments were quickly placed in seawater on a clean glass slide and a coverslip was added. Excess water was blotted from the edge and the whole mount was sealed with Valap (1:1:1 mixture of Vaseline, lanolin, and paraffin wax, melted at about 40 °C [36]. Within 2–5 min after wounding the slide was placed on the video microscopy system. Olympus BX51 research microscope (Olympus, Tokyo, Japan) equipped with fluorescence and differential interference contrast (DIC) optics with an oil-immersion condenser was used for photography. Single frame photographs were taken every 10 s for 5 days and all the frames were amalgamated into video clips using a time-lapse program (VideoVelocity, CandyLabs, Vancouver, BC, Canada).

### 2.3. Fluorescent and Histochemical Staining of ROS

Reactive oxygen species (ROS) was detected during the wound-healing process using 2′,7′-Dichlorofluorescin Diacetate (DCFH-DA, Sigma-Aldrich, St. Louis, MO, USA). After killing the cell and removing debris of cytoplasm, the wounded filaments were treated with DCFH-DA at 50 μM/mL and left in the dark for 5 min. After washing the cells with IMR media, fluorescence was observed under blue light with a narrow band blue filter. Green fluorescence was generated by reaction of DCFH-DA with O^2−^ [7].

Hydrogen peroxide (H_2_O_2_) was stained with 3,3′-diaminobenzidine (DAB). A reddish-brown precipitate is produced when DAB is oxidized by H_2_O_2_ in the presence of peroxidases [37,38]. The staining solutions were prepared by dissolving 1 mg DAB in 1 mL autoclaved sea water. Wounded filaments were immersed in this DAB staining solution. The tube was wrapped with aluminum foil to block light and kept at 10 °C for 12 h for staining. Therefore, DAB staining visualizes H_2_O_2_ accumulated in cells during the incubation period. To remove autofluorescence and enhance the staining, the filaments were washed twice with ethanol and transferred to microfuge tubes containing 100% methanol and incubated at 50 °C for 3–5 h, then observed with a light microscope.

### 2.4. Inhibition Experiment

Inhibition experiments were performed in a slide chamber filled with seawater. Upper and lower portions of a severed filament were treated with inhibitors and then placed in the slide chamber. Ascorbic acid was used to scavenge ROS. Diphenylene iodonium (DPI) was used to block the activity of NADPH-oxidases. DPI was pretreated to the filament for 30 min before wounding, while the other inhibitors were continuously applied during wound healing. Caffeine was used to block calcium influx through caffeine-sensitive Ca^2+^ channels [39,40,41,42]. As for the concentration of the inhibitors, the minimum dose required for inhibition obtained from preliminary experiments was used: 5 mM caffeine, 10 μM DPI, 0.5 mM ascorbic acid, and 50 μM H_2_O_2_, respectively. After inhibitor treatment, the effect was observed by placing the upper and lower part of the filaments obtained from different individuals in close proximity. The number of repair cells formed, growth of repair cells, and somatic cell fusion were examined over time.

### 2.5. RNA Extraction and cDNA Synthesis

Total RNA was extracted from *G. monilis* filaments over time during the wound-healing process. The RNA was extracted using TRIzol reagent (iNtRON Biotechnology Inc., Seongnam, Korea) and poly-A RNA was purified using an Oligotex mRNA purification kit (QIAGEN, Hilden, Germany). Quantification of total RNA was performed using NanoDrop ND-1000 spectrophotometer (NanoDrop Technologies, Wilmington, DE, USA) and quality-assessed by RNA 6000 Nano assay kit (Agilent Technologies, Palo Alto, CA, USA) and Bioanalyser2100 (Agilent Technologies, Palo Alto, CA, USA). Libraries were generated from one microgram of total RNA using TruSeq RNA Sample Prep Kit (Illumina, San Diego, CA, USA) according to the manufacturer’s protocol. After purification, the total poly A + RNA was fragmented into small pieces using divalent cations under elevated temperature. The cleaved mRNA fragments were reverse transcribed into first strand cDNA using random primers. Short fragments were purified with a QiaQuick PCR extraction kit and resolved with elution buffer for end reparation and addition of poly (A). Subsequently, the short fragments were connected with sequencing adapters. Each library was separated by adjoining distinct multiplexing indexing tag. All cDNAs were diluted to 10 μg/µL with autoclaved milli-Q water and stored at −20 °C.

### 2.6. 5′RACE PCR and Structure Analysis

Partial sequences of NADPH-oxidase homologues were obtained from our *G. monilis* transcriptome data. To obtain the full sequences, gene specific primer for *GmRBOH*s (Appendix A) was designed on the fragment to clone the full-length cDNA according to the RACE protocols. Geneious version 10 (www.geneious.com, accessed on 15 March 2022) was used to determine the primer site for RACE PCR. The first strand cDNA was synthesized using the SMARTer^®^ RACE 5′/3′ Kit (TaKaRa, Nojihigashi, Kusatsu, Shiga, Japan). cDNA samples were used to generate 5′ and 3′ cDNA fragments in PCR reactions combining them with SeqAmp DNA Polymerase and the gene specific primers. The PCR reaction was performed running 30 cycles of PCR Program described in Section VI of the SMARTer RACE 5′/3′ Kit User Manual. RACE products were then cloned into the linearized pRACE vector using the In-Fusion HD Cloning Kit included in the SMARTer RACE 5′/3′ Kit. The clones containing the largest gene-specific inserts were sequenced with M13 primers. The quality of the Sanger reads was analyzed manually. Full-length CDS of the protein were obtained. Nucleotide and amino acid sequence similarity searches and comparison were carried out using BLAST (blast.ncbi.nlm.nih.gov, accessed on 18 March 2022). Alignment of the amino acid sequence was performed in EMBnet (https://embnet.vital-it.ch, accessed on 19 March 2022).

### 2.7. Quantitative PCR and Statistical Analysis

To observe responses of homologues of NADPH-oxidase genes during the wound-healing response, real-time quantitative PCR (qPCR) was performed using QuantiSpeed SYBR Hi-Rox Kit (PhileKorea, Geumcheon-gu, Seoul, Korea) in a StepOnePlus Real-Time PCR System (Applied Biosystems, Waltham, MA, USA) as described previously [43]. cDNA samples were amplified in triplicate from the same RNA preparation. Wounded plants treated with different inhibitors were harvested at 1 h, 24 h, 48 h, and 96 h post wounding, and were frozen in liquid nitrogen. Unwounded plants were used as controls. RNA extraction and the synthesis of cDNA were performed using the same methods as described above. The final qPCR reaction volume was 20 μL and included 5 μL of diluted cDNA, 10 μL of iQ SYBR Green supermix and 200 nM of each primer (Appendix A). The reaction protocol was an initial 95 °C for 3 min and then 40 cycles of 95 °C for 15 s and 60 °C for 20 s. Relative expression levels were then calculated against the housekeeping gene EF1a as a calibrator using the delta-delta Ct method [44]. Statistical analysis was performed with GraphPad Prism 8 (GraphPad Software Inc., San Diego, CA, USA). Statistical significance was determined by two-way ANOVA with Bonferroni’s *post hoc* test for multiple comparisons. Significance was accepted at *p* < 0.05.

## 3. Results

### 3.1. Wound-Healing Process

When cells of *Griffithsia monilis* filaments were killed, repair rhizoidal cells (RRC) were first generated from the lower end of the upper adjacent cell within 24 h, and repair apical cell (RAC) was generated later from the upper part of the lower adjacent cell in 48 h (Figure 1A,B). Often two or more repair rhizoidal cells were formed sequentially from the upper adjacent cell (Figure 1C). The distal ends of both repair cells were filled with amorphous colorless materials (Appendix A, arrows). Repair rhizoidal cell became elongated and bent towards repair apical cell when the two repair cells came close to each other (Figure 1C, Appendix A). Both repair cells were attracted towards each other’s distal end, where somatic cell fusion took place (Figure 1D). Initially multiple repair rhizoidal cells grew towards the repair apical cell but once there was fusion between a RRC and RAC, the other RRC cells elongated away from the RAC (Appendix A).

### 3.2. Visualization of ROS during the Wound-Healing Process

Green fluorescence of ROS was observed by DCFH-DA staining in cells killed by physical damage (Figure 2A). ROS appeared to be released from the debris of degenerating intracellular organelles and was distinct from other autofluorescence in the cytoplasm (Figure 2(A2,A3), yellow arrows). ROS production in the adjacent cells was observed after removing the debris inside the dead cells (Figure 2B1). ROS were detected in the upper and lower adjacent cells facing the dead cell (Figure 2(B4,B5)), and also accumulated in the empty space of the dead cell 30 min post wounding (Figure 2B2). When adjacent cells produced repair cells, ROS production was confined to the repair cells (Figure 2C) and strong fluorescence was detected in the area close to repair apical cell (Figure 2C2). The strongest fluorescence was observed at the apical end of repair cells (Figure 2(C4,C5)).

As the two repair cells grew towards each other, ROS generation was localized at the distal end of each repair cell and fluorescence was mainly detected in the space where the two repair cells face each other (Figure 3A). When the two repair cells did not meet in the middle of the dead cell space, they bent towards each other’s distal end, and fluorescence was detected around the two repair cells (Figure 3B). When somatic fusion occurred between the two repair cells, ROS staining decreased and only traces were detected in the fusion region (Figure 3C). ROS fluorescence was not detected when fusion cells expanded, filling the space of dead cell 96 h after wounding (data not shown). DAB staining showed that ROS was released as hydrogen peroxide (H_2_O_2_) (Appendix A). A reddish-brown precipitate of oxidized DAB was observed at the distal end of both repair cells (Appendix A). When somatic cell fusion occurred, no specific staining was detected in the fusion cell or the empty space of the dead cell (Appendix A).

### 3.3. Effects of ROS and Calcium Inhibitors

In controls, somatic cell fusion occurred within 96 h, and the repair apical cells are in an extended form (Figure 1D). When cells were wounded in a solution containing 0.5 mM ascorbic acid, no repair apical cell was generated even by 96 h after wounding and the upper repair rhizoidal cell was not attracted to the lower adjacent cell (Figure 4A). When 0.5 mM ascorbic acid was applied after the formation of the repair cells, the attraction between the upper and lower repair cells disappeared and the repair apical cells became dome shape in 48 h (Figure 4B).

When the plants were pretreated with 10 μM DPI, an inhibitor of NADPH-oxidase, for 30 min before wounding, the wound response leading to somatic cell fusion was blocked (Figure 5). When DPI-pretreated cells and untreated cells were placed side by side, RRC as well as RAC only extended toward and fused with untreated cells (Figure 5(A1–A4,B1–B4)). ROS generation in adjacent and repair cells was also blocked with pretreatment of DPI (Appendix A). Almost no ROS was detected at the tip of repair cells with DCFH-DA staining (Appendix A).

Treatment with 5 mM caffeine reversibly blocked the generation of ROS as well as the wound response (Figure 6). When cells were wounded in the media containing 5 mM caffeine, no repair cell was generated in either the upper or lower adjacent cells by 72 h post wounding (Figure 6A1). However, the wound response resumed when the caffeine was washed out and somatic fusion between repair cells occurred in 96 h (Figure 6(A2–A4)). DCFH-DA staining showed that ROS generation also resumed when the caffeine was washed out (Figure 6(B1–B3)), but no more fluorescence was detected after somatic cell fusion (Figure 6B4).

Treatment with 50 μM H_2_O_2_ induced formation of multiple repair cells and somatic fusions (Appendix A). Numerous repair apical cells were generated from subapical region of the lower adjacent cell, which differs from controls in which only one or two repair apical cells were generated. As the number of repair cells participating in somatic fusion increased multiple connections were formed in various part of the lower adjacent cell (Appendix A).

### 3.4. NADPH-Oxidase Homologues and Their Expression in Response to Wounding

The fragmentary cDNA sequences of NADPH-oxidase homologues were obtained from our *G. monilis* transcriptome data. Based on the transcriptome data each gene specific primers (GSP) were designed and used for 5′ and 3′ RACE PCR of the cDNA. Full-length CDS sequences with different sizes (3658, 2844, 2433, and 2550 bp) were obtained and named as *GmRBOH* 1–4, respectively (Figure 7, Appendix A). Active domains of NADPH-oxidase were well conserved in all *GmRBOH*s and they showed similar secondary structures in having ten transmembrane helices, with attachment sites for two noncovalent heme molecules and a FAD domain close to the N-terminal region and NAD-binding domains at rear region (Figure 7, Appendix A). Typically, *GmRBOH*s lack an EF-hand domain which serves as a calcium binding site (Appendix A). The *Gmrboh* genes showed low (20.2–35.1%) amino acid sequence identity to each other.

Quantitative PCR analysis showed that the expression of *GmRBOH*s was highly upregulated by the wounding and gradually reduced during wound healing (Figure 8, Orange bars). This expression pattern was observed in all *GmRBOH*s, but the fold change in expression varied from >3 to >62. The expression of *GmRBOH**2* and *GmRBOH**3* increased most significantly, between >11.3 and >62.3, respectively (Figure 8B,C). When the plants were treated with 5 mM caffeine, the expression of *GmRBOH*s was initially lower than that of the wounded plants, 1 h after wounding, but gradually increased over time to be higher than that of the wounded plants (Figure 8, grey bars). In the treatment with 0.5 mM ascorbic acid and 50 μM H_2_O_2_, wounding of plants did not induce upregulation of any *GmRBOH*s (Figure 8, yellow and pale blue bars, respectively). When plants were preincubated in 10 μM DPI for 30 min before wounding, the expression of *GmRBOH*s was not upregulated upon wounding and did not recover over time (Figure 8, green bars).

## 4. Discussion

Our results showed that a calcium-mediated ROS signal regulates the wound healing response in *Griffithsia monilis*, from generation and elongation of the repair cells to the fusion process. Early wound recognition by adjacent cells appears to be triggered by ROS released from degenerating dead cells because the wound response did not initiate when cells were treated with a ROS scavenger. After this step, ROS production appears to be driven by NADPH-oxidases localized to adjacent cells as well as repair cells involved in wound healing since ROS generation stopped with treatment of DPI or caffeine, inhibitors of NADPH-oxidase (Figure 9). The production, local concentration, and exposure time of ROS needs to be tightly controlled, as it can be involved in multiple intracellular reactions and may lead to a variety of toxic responses [45]. Our results showed that ROS generation was mostly restricted to adjacent cells facing the dead cells, and shifted to the developing repair cells, and then to their elongated tips. ROS are released from these cell regions into the extracellular space as H_2_O_2_ and ROS generation stopped as soon as somatic cell fusion occurred. A positive feedback loop of H_2_O_2_ and NADPH-oxidase activity was observed. Ascorbic acid treatment not only eliminates the released H_2_O_2_ and interferes with signal transduction, but also inhibits the expression of the NADPH-oxidase gene in the early stages of wound healing (Figure 9). Quantitative PCR results also showed that expression of *GmRBOH*s was inhibited when NADPH-oxidase was inactivated by DPI or caffeine treatment. Hydrogen peroxide appears to be a direct signal that induces cellular changes. H_2_O_2_ treatment induced numerous repair cells in the area, the subapical region of lower adjacent cells, where they are not normally produced.

Animals and plants have evolved sophisticated mechanisms that regulate their responses to mechanical injury, and ROS play a pivotal role in the orchestration of wound-healing process [2,3]. ROS has several advantages as a signaling molecule for the wound response. Large amounts of ROS are automatically generated during the degeneration of organelles following mechanical damage to cells [46]. The ability of cells to rapidly produce ROS simply by balancing the rates of cellular production and scavenging allows the organism to produce ROS with little cost [47]. As a rapid auto-propagating signal, ROS could activate NADPH-oxidases in cells along the path of the signal, which amplifies the total amount of ROS produced at the wound site quickly. As a bacteriostatic agent, ROS kills pathogens present in the wound, resulting in more ROS leakage into the wound site [3]. Membrane permeability is essential for a signal molecule in the wound response. Hydrogen peroxide (H_2_O_2_) can across membranes of target cells, either passively or through water channels [48]. It is therefore not surprising that wound responses in animals and plants are largely dependent on intercellular communication regulated by H_2_O_2_ [2,46]. Our results suggest that ROS generated upon cell injury may be an initial signal for the wound response, but a continuous supply of ROS driven by NADPH-oxidases in repair cells is essential to sustain the wound-healing process as it not only induces differentiation of the repair cells, but also mediates directional elongation of repair cells up to somatic cell fusion.

Repair cells in *G. monilis* elongated and bend towards higher ROS (probably H_2_O_2_) concentration as shown in DCFH-DA staining. When repair cells grow along the H_2_O_2_ gradient, the tips of the two repair cells will eventually meet as the highest concentration of H_2_O_2_ is at the tips of each repair cell. H_2_O_2_ can induce cell wall loosening by generating hydroxyl radicals which cleave polysaccharide bonds in the cell wall [49], promoting expansion of cells and facilitating fusion of repair cells. Oxidation of cell wall polysaccharides by hydrogen peroxide has long been proposed as a potential mechanism for cell wall breakdown in plants [50], but studies have also shown that H_2_O_2_ mediates cell-wall stiffening by cross-linking of phenolic compounds in cell wall [51]. Although the effect of hydrogen peroxide on loosening of cell wall varies with the cell type and wall composition, it is generally believed that interaction between H_2_O_2_ and hormones play an important role in the elongation of plant cells [52]. For example, auxin participates in cell growth by inducing cell wall peroxidases (peroxidases class III) and NADPH oxidases to produce ROS and promote cell wall loosening and further cell elongation [53]. It is interesting that the direction of repair cell elongation could be determined simply by restricting the sites from which hydrogen peroxide is released during the wound response in *Griffithsia monilis*.

Rhodomorphin has been proposed as a hormone that induces differentiation of repair apical cells and its elongation towards repair rhizoidal cells in *Griffithsia pacifica* [26,32]. The presence of rhodomorphin was only indirectly shown by tracking the cellular reaction to extracts from wounded plants or by observing the secretion of concanavalin A-specific proteins [29,33,34]. None of the above studies used purified protein to induce differentiation or elongation of repair cells. Waaland and Watson (1980) reported that rhodomorphin is only released from repair rhizoidal cells and mediates differentiation of repair apical cell, and not vice versa. However, our results showed that both repair cells produced ROS at their distal ends and were able to induce the elongation of complementary repair cells towards them. H_2_O_2_ treatment induced numerous repair apical cells in areas where repair apical cells were not normally produced, and ascorbic acid treatment reversed this differentiation. The role of rhodomorphin is therefore called into question, and these experiments need to be investigated again.

Calcium influx through caffeine-sensitive channel appears to be an upstream signal that regulates wound responses, including initial activation of NADPH-oxidases and ROS production as well as their gene expression. Caffeine treatment may block the generation of repair rhizoidal cells which were not affected by other inhibitors, and the wound response, including ROS production, resumes as soon as the caffeine was washed away. In brown algae, the role of H_2_O_2_ in maintaining Ca^2+^ gradient has been well documented during embryo development [17,18]. A positive feedback loop between H_2_O_2_ and Ca^2+^ signal modulates ROS level in Fucus zygotes during their development [19]. Recently, a calcium-mediated H_2_O_2_ signal has been reported to regulate fertilization in the red alga, *Bostrychia moritziana* [7]. The interaction between calcium and H_2_O_2_ signal during the fertilization process is different from that of wound response in *Griffithsia monilis*. H_2_O_2_ treatment restored cell cycle progression in spermatia even when the spermatia were treated with caffeine [7], while the wound response in *Griffithsia monilis* did not resume with H_2_O_2_ treatment alone. Which of the calcium and ROS signals takes precedence seems to depend on the cellular and developmental context.

NADPH-oxidases are involved in ROS generation in most kingdoms of life [9]. NADPH-oxidases share a catalytic core formed by six transmembrane helices, with two noncovalent heme molecules, followed by a C-terminal dehydrogenase domain that binds NADPH and FAD [54]. These structures are well conserved in identified *GmRBOH*s. Four *GmRBOH*s displayed similar expression patterns during wound healing, but with different expression levels over time in response to inhibitor treatment. The expression of *GmRBOH*s during the wound response required Ca^2+^ influx along with a H_2_O_2_ signal, but each *GmRBOH* responds differently to caffeine treatment in expression. *GmRBOH*s do not have an EF domain essential for calcium binding, and little is known about the calcium signaling pathways in red algae [7]. In plants, some NADPH-oxidases have calcium-binding EF-domains but their ROS-producing catalytic activity as well as the structural basis of the enzyme regulation by calcium remains mostly unknown [54]. In red algae, several NADPH-oxidases were identified and their involvement in various cellular response has been described [13,14,55], but all of the NADPH-oxidases reported so far lacks EF-domains [7,20]. Further studies on the interaction of NADPH-oxidases and the enzymes mediating calcium-signaling are necessary to understand the role of ROS in regulating cellular responses in red algae.

A key signaling advantage of ROS is their tight link to cellular homeostasis and metabolism [56]. Fatal wounding of cells causes destruction of intracellular organelles and inevitably leads to the release of large amounts of ROS, changing the steady-state level of ROS in adjacent cells. Our results show that this ROS signal, and calcium influx, activate NADPH-oxidases of the cells involved in wound healing to generate more ROS, leading to upregulation of *GmRBOH*s in a positive feedback loop. By limiting the intracellular sites where ROS is generated, repair cells grow along the gradient of released H_2_O_2_, naturally meet each other, undergo somatic cell fusion, and successfully restore the multicellular system. Further studies on the components and pathways of calcium signaling involved in the activity of NADPH-oxidases are essential to understand the unique ROS signaling networks in red algae.

## Figures and Tables

**Figure 1 cells-11-02124-f001:**
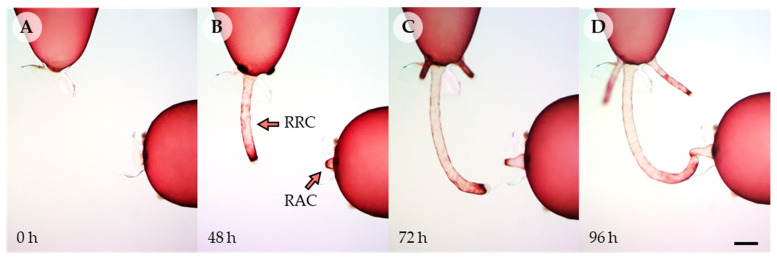
Induced elongation and somatic fusion of repair cells in *Griffithsia monilis*. (**A**) After removal of dead cell debris, two adjacent cells were placed close to each other. (**B**) Repair rhizoidal cell and repair apical cell were generated and extended from upper and lower adjacent cells, respectively. (**C**) The first repair rhizoidal cell began to bend towards the repair apical cell. The second and third repair rhizoidal cell formed from the upper adjacent cell began to elongate. (**D**) Somatic fusion occurred at the ends of the two repair cells. RRC; repair rhizoidal cell, RAC; repair apical cell. Scale bar: 200 µm.

**Figure 2 cells-11-02124-f002:**
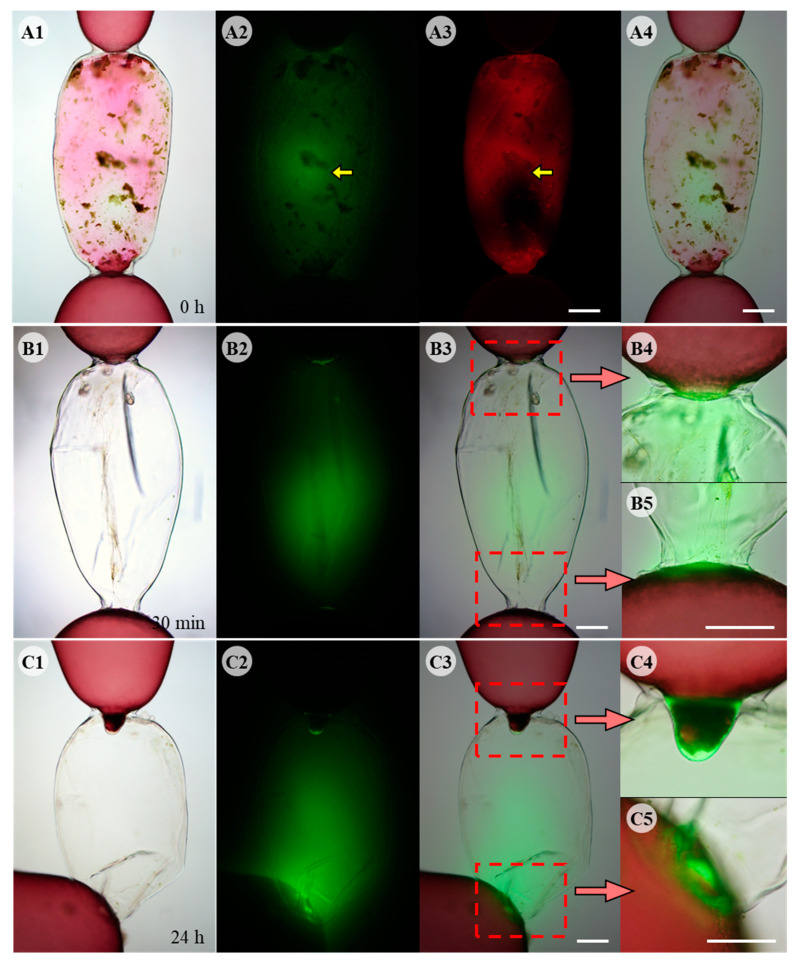
Fluorescent DCFH-DA staining during wound healing in *Griffithsia monilis*. (**A1**) Light microscope image of physically damaged cell showing cellular debris. (**A2**) DCFH-DA staining image showing ROS inside the same cell. (**A3**) Autofluorescence of released pigments of dead cell observed with different filter. (**A4**) Merged image of (**A1**,**A2**) showing ROS inside dead cell. (**B1**) Light microscope image of damaged cell 30 min after removal of the debris. (**B2**) DCFH-DA staining showing ROS inside the same cell. (**B3**) Merged image of (**B1**,**B2**). (**B4**) Enlarged image of (**B3**) showing upper adjacent cell. (**B5**) Enlarged image showing lower adjacent cell. (**C1**) Light microscope image of developing repair cells. (**C2**) DCFH-DA staining showing ROS generated from the tips of repair cells filling the dead cell space. (**C3**) Merged image. (**C4**) Enlarged image of repair rhizoidal cell. (**C5**) Enlarged image of developing repair apical cell. Scale bars: 200 µm.

**Figure 3 cells-11-02124-f003:**
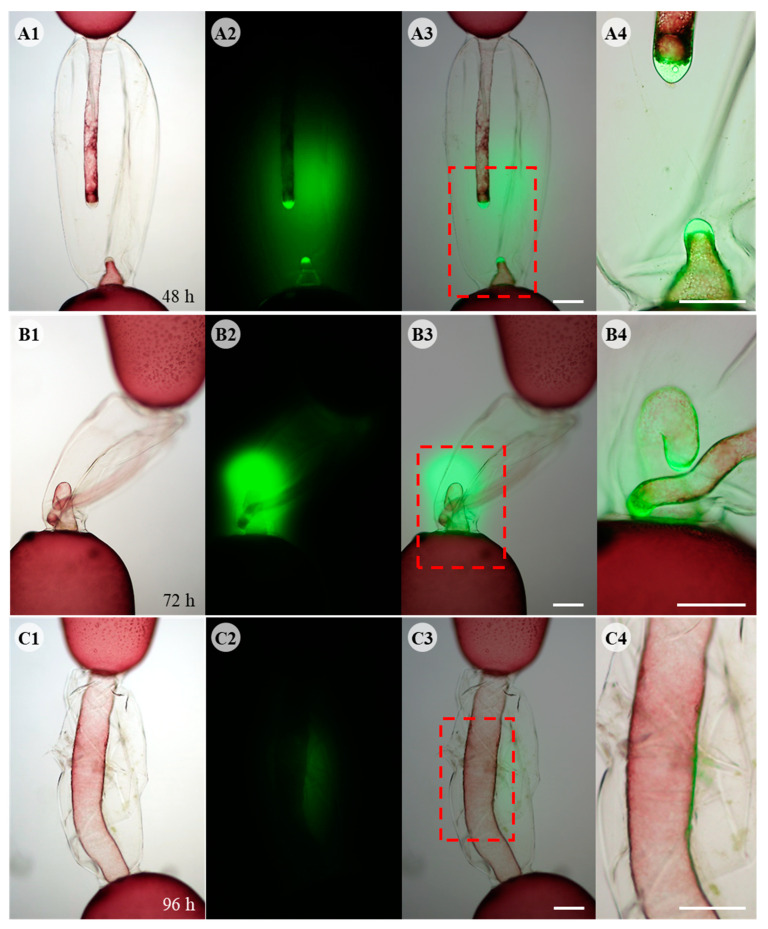
Fluorescent DCFH-DA staining during wound healing in *Griffithsia monilis*. 1st column shows light microscope images, 2nd column shows DCFH-DA fluorescence image of the same cell, 3rd column shows the merged images, and 4th column shows enlarged image of dashed box in 3rd column. (**A1**) Light microscope image showing repair cells growing towards each other. (**A2**) DCFH-DA staining image of the same cell showing ROS production at the tip of each repair cell and strong fluorescence between the two repair cells. (**A3**) Merged image of (**A1**,**A2**). (**A4**) Enlarged image of the dashed box in (**A3**). (**B1**) Light microscope image of repair cells just before somatic fusion. Repair apical cell bent towards repair rhizoidal cell. (**B2**) DCFH-DA staining showing strong fluorescence in the contact area around the two repair cells. (**B3**) Merged image. (**B4**) Enlarged image of dashed box. The two repair cells are in contact with each other, but somatic fusion does not occur until the distal ends of each other are in contact. (**C1**) Light microscope image of fused cell. (**C2**) DCFH-DA staining disappeared except for faint fluorescence around fusion area. (**C3**) Merged image. (**C4**) Enlarged image of dashed box. Scale bars: 200 µm.

**Figure 4 cells-11-02124-f004:**
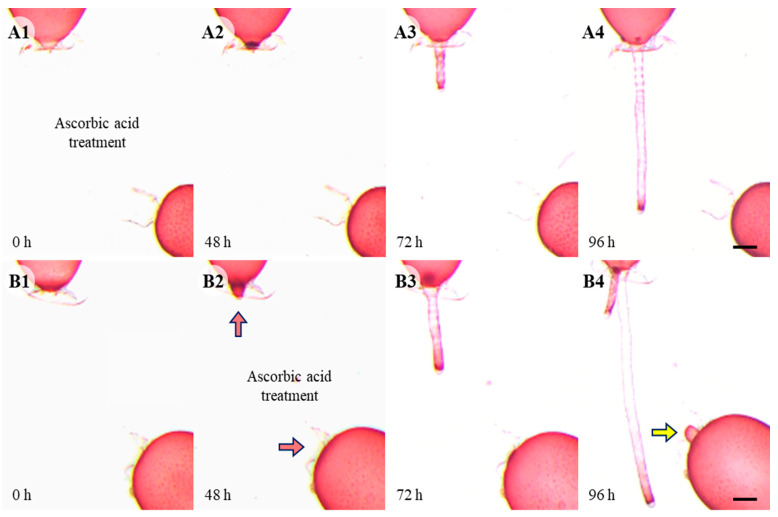
Wound healing with ascorbic acid treatment. (**A1**–**A4**) Repair apical cell was not produced even by 96 h. Repair rhizoidal cells developed later than control and was not attracted to lower cell. (**B1**–**B4**) When ascorbic acid was treated after the development of repair cells (48 h, red arrow), repair cells are not attracted towards each other. Yellow arrow points normal cell not involved in fusion, not repair cell. Scale bars: 200 µm.

**Figure 5 cells-11-02124-f005:**
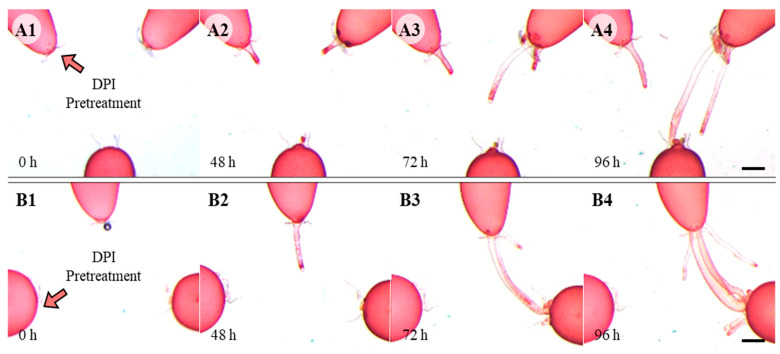
Wound healing with pretreatment of DPI for 30 min. After placing DPI-pretreated cells and non-treated cells close to the complementary repair cells, the attraction between the repair cells was observed. (**A1**–**A4**) When one of the upper cells was pretreated with DPI only the repair rhizoidal cells formed from untreated cells were attracted towards and fuse with repair apical cell. (**B1**–**B4**) When one of the lower adjacent cells was pretreated with DPI, the repair rhizoidal cell was attracted only towards untreated repair apical cell. Scale bars: 200 µm.

**Figure 6 cells-11-02124-f006:**
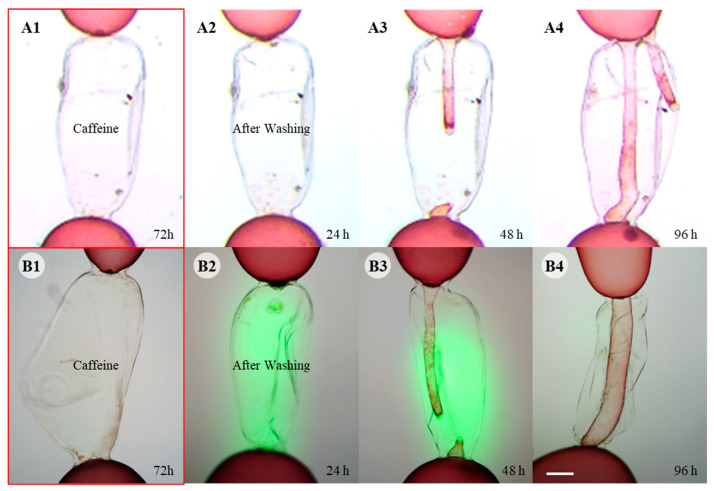
Wound healing with caffeine treatment. (**A1**). No repair cell was formed in caffeine treatment for 72 h. (**A2**–**A4**) Wound healing resumed after washing out the caffeine and somatic cell fusion occurred in 96 h after washing. (**B1**) DCFH-DA staining shows no fluorescence in caffeine treated cells for 72 h. (**B2**) Fluorescence was detected in 24 h after caffeine was washed out. (**B3**) Repair cells developed within 48 h after washing out caffeine, and strong fluorescence was detected at the tip of the elongated repair cells. (**B4**) Somatic cell fusion occurred in 96 h after washing out caffeine. No fluorescence was detected at this time. Scale bar: 200 µm.

**Figure 7 cells-11-02124-f007:**
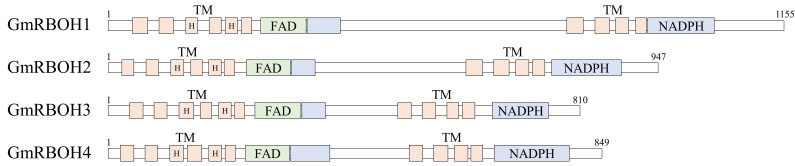
Functional domains of four *GmRBOH* genes (1, 2, 3, 4) identified in *Griffithsia monilis*. The number and order of domains are similar among *GmRBOH*s.

**Figure 8 cells-11-02124-f008:**
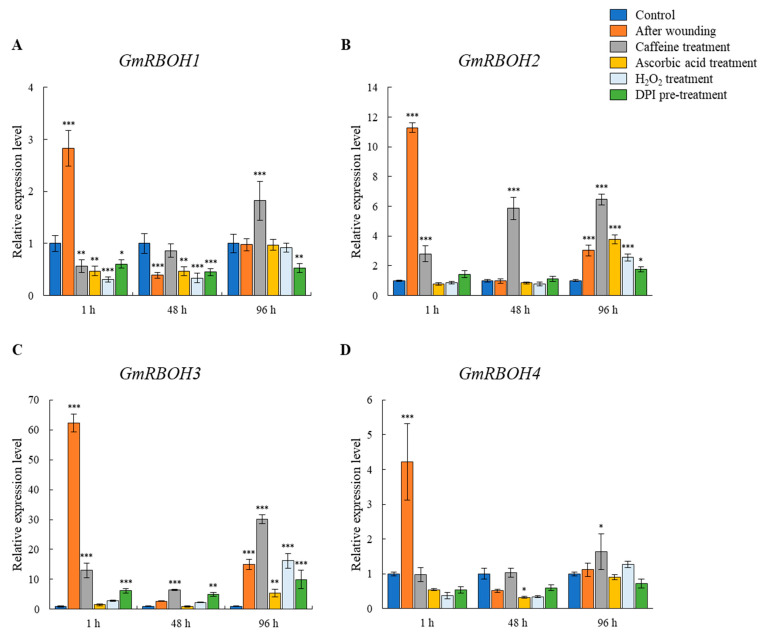
Expression of four *GmRBOH*s during wound healing in *Griffithsia monilis*. Upregulation of *GmRBOH*s was induced by injury in 1 h. Ascorbic acid, DPI and H_2_O_2_ treatments suppressed the expression of *GmRBOH*s upon wounding, whereas the expression of *GmRBOH*s was less suppressed by caffeine treatment. The blue bar shows expression in control. The orange bar represents the expression when plants were wounded. The grey bar shows *GmRBOH*s expression after caffeine treatment. The yellow bar shows expression in the ascorbic acid treatment. The pale blue bar shows expression in H_2_O_2_ treatment. The green bar shows the expression after DPI pretreatment. (**A**) Expression of *GmRBOH1* peaks 1 h after injury and decreases over time. (**B**) Expression of *GmRBOH2* increases 1 h after wounding and then decreases. *GmRBOH2* expression in caffeine treatment recovered over time. (**C**) Expression of *GmRBOH3* increased over 60 times more than that in control in 1 h after wounding. *GmRBOH3* expression in caffeine treatment remained higher than that of control. (**D**) The expression of *GmRBOH4* showed similar pattern to that of *GmRBOH1*. Data are expressed as mean ± SD (two-way ANOVA vs. control with Bonferroni’s multiple comparisons test, *n* = 3). * *p* < 0.05, ** *p* < 0.01, *** *p* < 0.001.

**Figure 9 cells-11-02124-f009:**
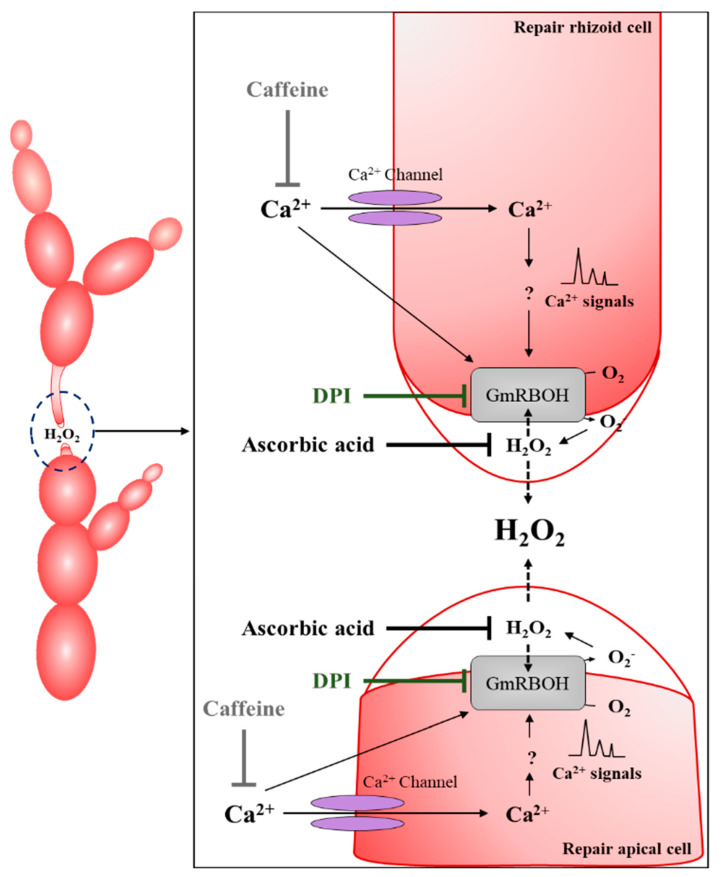
Diagram summarizing the calcium and H_2_O_2_ signaling and the effect of inhibitors during the wound healing response in *Griffithsia monilis.* Calcium influx through a caffeine−sensitive channel activates *GmRBOH*s, and the ROS produced by *GmRBOH*s are released into the extracellular space as H_2_O_2_ to activate *GmRBOH*s in other repair cells. DPI pretreatment disrupts the above positive feedback loop by blocking the activation of *GmRBOH*s. Ascorbic acid treatment has the same effect by scavenging the H_2_O_2_. Caffeine treatment inhibits the activation of *GmRBOH*s and inhibits wound healing.

## Data Availability

All data are contained within this article or Appendix A.

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
