# Peer review of "ROS Signaling Mediates Directional Cell Elongation and Somatic Cell Fusion in the Red Alga Griffithsia monilis"

_cells, 2022, doi:10.3390/cells11132124_

Round 1

Reviewer 1 Report

In this article, the authors aim to elucidate the key signals that trigger the initial wound-response, chemicals mediating the directional elongation of repair cells leading to somatic fusion, and the key genes involved in this process. Though time-lapse observation of wound response, fluorescent and histochemical staining of ROS and inhibition experiment, the authors found that ROS released from injured cell is an initial trigger of repair cell division, and subsequently ROS generated from repair cells are released externally as H2O2 to mediate repair cell elongation and somatic fusion in Griffithsia monilis. Repair cells elongate along the H2O2 concentration, meet at each other’s tips and undergo somatic fusion. Furthermore, four G. monilis homologues of NADPH-oxidase (Gmrbohs) were identified using genetic analysis. In conclusion, the study suggests that ROS generated upon cell injury activates Ca2+ channels and upregulates the expression of Gmrbohs, and that H2O2 generated from repair cells mediates induced repair cell elongation leading to somatic fusion and filament repair. Experimental design and results in this investigation are quite persuasive.  This is a carefully done study and the findings are of considerable interesting, a few minor revisions are listed below:

1、  The table style in the article must be transformed into table styles specified in the paper.

2、  The authors should analyze the significance between different groups in the investigation, and mark the number of repetitions of the experiment.

3、  There are some formatting and words expressing problems should be corrected, such as “95 ℃” in page 4, line 154, 155, 172.

4、  The abstract should include the experimental method used in the investigation.

Author Response

1. The table style in the article must be transformed into table styles specified in the paper.

A: The table style has been changed as suggested.

2. The authors should analyze the significance between different groups in the investigation, and mark the number of repetitions of the experiment.

A: We agree. The statistical analysis of the qPCR results have been incorporated in the figure and the results. We also revised M&M section to include statistical methods.

3、There are some formatting and words expressing problems should be corrected, such as “95 ℃” in page 4, line 154, 155, 172.

   A: It has been corrected.

4、 The abstract should include the experimental method used in the investigation.

   A: As suggested, a brief sentence mentioning the experimental method has been incorporated in the abstract

Reviewer 2 Report

Abstract:

Please integrate better the results you are presenting, it looks like a series of sentences without connections.

Introduction:

- Review the English from the line 24 to 30;

- line 30-31, "The signaling effects of ROS depends on the cellular context, 30 both its local concentration as well as its exposure time": review this sentence, its meaning is not clear ;

Methods:

2.4. Inhibition experiment

line 122-124: " DPI was pretreated to the filament for 30 min before wounding while other inhibitors were continuously treated during wound healing": re-write better this sentence, the grammar is wrong and difficult to understand the real meaning.

In addition, please report in supplementary data the preliminary experiments you have done to set the inhibitor's concentration to use in the final experiment.

2.7. Quantitative PCR

please add results normalized with at least two reference genes, has it is suggested in the Shim et al 2016 paper.

2.6. 5’RACE PCR and structure analysis

- please add results of this experiment in the main text.

- "Full-length sequences and the open reading frame (ORF) of the protein were obtained as described previously [39]" I cannot find this method in the citated article. please check it .

Discussion

- line 331-332: "Our results showed that a calcium-mediated ROS signal regulates the wound healing 331 response in Griffithsia monilis, from generation and elongation of the repair cells to fusion process": I don't think you can make this asumption as you don't have made any experiments demonstrating calcium involvment;

- line 345-347: "Ascorbic acid treatment not only eliminates the released H2O2 and interfering with signal transduction, but also inhibited the ROS production itself by suppressing the expression of the NADPH oxidase gene": make grammatical corrections. in addition, how you can be sure? the qPCR result don't show always a reduction of expression in respect to the control when the treatments are made. In addition not always the gene expression is correlated with the correspondin protein activity. You should show an inhibition of the protein enzymatic activity and a western blot showing  a reduction in the amount of the correspondin protein. Please provide this results or re-modulate the sentence.

Author Response

Abstract:

Please integrate better the results you are presenting, it looks like a series of sentences without connections.

A: We have revised the Abstract as suggested

Introduction:

- Review the English from the line 24 to 30;

A: English has been revised as suggested.

- line 30-31, "The signaling effects of ROS depends on the cellular context, 30 both its local concentration as well as its exposure time": review this sentence, its meaning is not clear ;

A: The sentence has been changed to convey the meaning more clearly.

Methods:

2.4. Inhibition experiment

line 122-124: " DPI was pretreated to the filament for 30 min before wounding while other inhibitors were continuously treated during wound healing": re-write better this sentence, the grammar is wrong and difficult to understand the real meaning.

A: The sentence has been re-written for better understanding

In addition, please report in supplementary data the preliminary experiments you have done to set the inhibitor's concentration to use in the final experiment.

   A: The results of preliminary experiments to set the inhibitor’s concentration has been incorporated as a new Supplementary Figure 4.

2.7. Quantitative PCR

Please add results normalized with at least two reference genes, has it is suggested in the Shim et al 2016 paper.

   A: We tried different genes to determine the appropriate reference gene (Please see figures attached). As EF1 was the only gene unaffected by experimental factors and meet the criteria described by Shim et al. (2016) we did not add other reference genes.

2.6. 5’RACE PCR and structure analysis

Please add results of this experiment in the main text.

A: We incorporated the results of 5’RACE PCR in the results as suggested.

"Full-length sequences and the open reading frame (ORF) of the protein were obtained as described previously [39]" I cannot find this method in the citated article. please check it .

A: We corrected the reference as suggested.

Discussion

- line 331-332: "Our results showed that a calcium-mediated ROS signal regulates the wound healing response in Griffithsia monilis, from generation and elongation of the repair cells to fusion process": I don't think you can make this assumption as you don't have made any experiments demonstrating calcium involvement;

A: We believe that inhibition experiments with caffeine support this assumption. Caffeine has been used as a specific calcium inhibitor in many studies. It has long been generally accepted that calcium signaling is involved in cellular responses when caffeine inhibits calcium flow into cells. Please see following articles that mentioned it.

Sorenson, Coelho & Reuben. 1986. Caffeine inhibition of calcium accumulation by the sarcoplasmic reticulum in mammalian skinned fibers. J. Membr, Biol. 90: 219-230.

Zholos & Shuba 1991. The inhibitory action of caffeine on calcium currents in isolated intestinal smooth muscle cells. Pflugers Arch. 419: 267-73

Kang et al. 2010. Caffeine-Mediated Inhibition of calcium release channel inositol 1,4,5-Trisphosphate Receptor Subtype 3 Blocks Glioblastoma Invasion and Extends Survival. Cancer Research. doi: 10.1158/0008-5472.CAN-09-2886

- line 345-347: "Ascorbic acid treatment not only eliminates the released H2O2 and interfering with signal transduction, but also inhibited the ROS production itself by suppressing the expression of the NADPH oxidase gene": make grammatical corrections. in addition, how you can be sure? the qPCR result don't show always a reduction of expression in respect to the control when the treatments are made. In addition not always the gene expression is correlated with the correspondin protein activity. You should show an inhibition of the protein enzymatic activity and a western blot showing a reduction in the amount of the correspondin protein. Please provide this results or re-modulate the sentence.

A: As suggested, the sentence has been modified to convey the meaning more clearly.

Round 2

Reviewer 2 Report

Abstract: please introduce the species for which you are describing the results. You never mention it unless at the end and with an abbreviation.

Introduction: in the introduction is completely missing any information about the species subject of the experiments and why you are using that specie. Please describe the specie in the ocntext of the experimentation done and justify its use as model in this context.

Results:

- line 305: "Based on the sequence obtained 3` rapid amplification of cDNA ends"

what does it means?

2.6. 5’RACE PCR and structure analysis

Please add results of this experiment in the main text.

A: We incorporated the results of 5’RACE PCR in the results as suggested. 

where ? I cannot find any results and figure regardin it in the main text.

Previous comments and answer:

 line 331-332: "Our results showed that a calcium-mediated ROS signal regulates the wound healing response in Griffithsia monilis, from generation and elongation of the repair cells to fusion process": I don't think you can make this assumption as you don't have made any experiments demonstrating calcium involvement;

A: We believe that inhibition experiments with caffeine support this assumption. Caffeine has been used as a specific calcium inhibitor in many studies. It has long been generally accepted that calcium signaling is involved in cellular responses when caffeine inhibits calcium flow into cells. Please see following articles that mentioned it.

Sorenson, Coelho & Reuben. 1986. Caffeine inhibition of calcium accumulation by the sarcoplasmic reticulum in mammalian skinned fibers. J. Membr, Biol. 90: 219-230.

Zholos & Shuba 1991. The inhibitory action of caffeine on calcium currents in isolated intestinal smooth muscle cells. Pflugers Arch. 419: 267-73

Kang et al. 2010. Caffeine-Mediated Inhibition of calcium release channel inositol 1,4,5-Trisphosphate Receptor Subtype 3 Blocks Glioblastoma Invasion and Extends Survival. Cancer Research. doi: 10.1158/0008-5472.CAN-09-2886

these are all papers in mammalian cells, but plants can be very different so you have to demonstrate it without any assumption derived from animals comparison

Author Response

Comments and Suggestions for Authors

Abstract:

Please introduce the species for which you are describing the results. You never mention it unless at the end and with an abbreviation.

A: We incorporated the species name as suggested

Introduction:

In the introduction is completely missing any information about the species subject of the experiments and why you are using that species. Please describe the specie in the context of the experimentation done and justify its use as model in this context.

A: We incorporated information about the species as suggested.

Results:

- 2.6. 5’RACE PCR and structure analysis

Please add results of this experiment in the main text.

A: We have revised M&M and Results to incorporate the procedures and results of the experiment. A new figure describing the results has been added in the main text and Figure S3.

Previous comments and answer:

 … these are all papers in mammalian cells, but plants can be very different so you have to demonstrate it without any assumption derived from animals comparison

A: We agree. Please see following articles mentioning about the caffeine and calcium relationships in plants and algae.

Länge, S., 1, Wissmann, J.D., Plattner, H. 1996. Caffeine inhibits Ca2+ uptake by subplasmalemmal calcium stores ('alveolar sacs') isolated from Paramecium cells. Biochim Biophys Acta. 31;1278(2):191-6.

Yasuhara H. 2005. Caffeine inhibits callose deposition in the cell plate and the depolymerization of microtubules in the central region of the phragmoplast. Plant cell physiology. Plant Cell Physiol. 46(7): 1083–1092.

Yeung P. et al. 2006. Involvement of calcium mobilization from caffeine-sensitive stores in mechanically induced cell cycle arrest in the dinoflagellate Crypthecodinium cohnii. Cell calcium 39: 259-74.

Round 3

Reviewer 2 Report

Previous comments and answer:

… these are all papers in mammalian cells, but plants can be very different so you have to demonstrate it without any assumption derived from animals comparison

A: We agree. Please see following articles mentioning about the caffeine and calcium relationships in plants and algae.

Länge, S., 1, Wissmann, J.D., Plattner, H. 1996. Caffeine inhibits Ca2+ uptake by subplasmalemmal calcium stores ('alveolar sacs') isolated from Paramecium cells. Biochim Biophys Acta. 31;1278(2):191-6.

Yasuhara H. 2005. Caffeine inhibits callose deposition in the cell plate and the depolymerization of microtubules in the central region of the phragmoplast. Plant cell physiology. Plant Cell Physiol. 46(7): 1083–1092.

Yeung P. et al. 2006. Involvement of calcium mobilization from caffeine-sensitive stores in mechanically induced cell cycle arrest in the dinoflagellate Crypthecodinium cohnii. Cell calcium 39: 259-74.

please cite these papers in the text